# Metrics based on information entropy applied to evaluate complexity of landscape patterns

**Sérgio Henrique Vannucchi Leme de Mattos**[1]*, **Luiz Eduardo Vicente**[2], **Andrea Koga Vicente**[2], **Cláudio Bielenki Júnior**[1], **José Roberto Castilho Piqueira**[3]

1 Environmental Complex Systems Laboratory, Department of Hydrobiology, Biological and Health Sciences Center, Federal University of São Carlos (UFSCar), São Carlos, Brazil, 2 ABC Platform—Embrapa Environment, Jaguariúna, Brazil, 3 Department of Telecommunications and Control Engineering, Polytechnic School of the University of Sao Paulo (USP), São Paulo, Brazil

☯ These authors contributed equally to this work.
* sergiomattos@ufscar.br

**Data Availability Statement:** All relevant data will be held in https://github.com/lascaufscar.

**Funding:** The authors received no specific funding for this work.

## Abstract

Landscape is an ecological category represented by a complex system formed by interactions between society and nature. Spatial patterns of different land uses present in a landscape reveal past and present processes responsible for its dynamics and organisation. Measuring the complexity of these patterns (in the sense of their spatial heterogeneity) allows us to evaluate the integrity and resilience of these complex environmental systems. Here, we show how landscape metrics based on information entropy can be applied to evaluate the complexity (in the sense of spatial heterogeneity) of patches patterns, as well as their transition zones, present in a Cerrado conservation area and its surroundings, located in south-eastern Brazil. The analysis in this study aimed to elucidate how changes in land use and the consequent fragmentation affect the complexity of the landscape. The scripts *CompPlex HeROI* and *CompPlex Janus* were created to allow calculation of information entropy ($H_e$), variability ($H_e/H_{max}$), and López-Ruiz, Mancini, and Calbet (LMC) and Shiner, Davison, and Landsberg (SDL) measures. *CompPlex HeROI* enabled the calculation of these measures for different regions of interest (ROIs) selected in a satellite image of the study area, followed by comparison of the complexity of their patterns, in addition to enabling the generation of complexity signatures for each ROI. *CompPlex Janus* made it possible to spatialise the results for these four measures in landscape complexity maps. As expected, both for the complexity patterns evaluated by CompPlex HeROI and the complexity maps generated by CompPlex Janus, the areas with vegetation located in a region of intermediate spatial heterogeneity had lower values for the He and He/Hmax measures and higher values for the LMC and SDL measurements. So, these landscape metrics were able to capture the behaviour of the patterns of different types of land use present in the study area, bringing together uses linked to vegetation with increased canopy coverage and differentiating them from urban areas and transition areas that mix different uses. Thus, the algorithms implemented in these scripts were demonstrated to be robust and capable of measuring the variability in information levels from the landscape, not only in terms of spatial datasets but also spectrally. The automation of measurement calculations, owing to informational entropy

**Competing interests:** The authors have declared that no competing interests exist.

provided by these scripts, allows a quick assessment of the complexity of patterns present in a landscape, and thus, generates indicators of landscape integrity and resilience.

## Introduction

Landscape is a level of ecological organisation comprising 'natural' and/or more anthropised ecosystems and is characterised as a complex environmental system. These interrelated and interdependent units form heterogeneous spatial mosaic. As spatial processes are intrinsically complex [1], especially when they involve relations between society and nature, studies on landscape ecology require a worldview and scientific paradigm, which comprehend such complexity. Such complexity worldview requires focus on understanding the interconnections and historical and contextualised processes that generate a diversity of forms and patterns [2], i.e., a view which must be supported by the principles and methods encompassed by the complexity paradigm.

Landscape complexity is directly related to these spatial and temporal heterogeneities. As noted by Fabrig and Nuttle [3], spatial heterogeneity can be divided into two non-excluding components: 1) compositional heterogeneity, related to different cover types; 2) configurational heterogeneity, associated with spatial pattern. Additionally, these spatial and temporal complexities in a landscape can be perceived not only inside each patch and among patches, but also at the intersections of these units.

Boundaries among these landscape units are not always well defined, as there is often a transition gradient between them. This transition area is called the ecotone, a concept used by Clements in 1905 to characterise a region where there are overlaps and interactions between two or more adjacent communities [4]. This gradient may be caused by natural factors, such as correlated differences between vegetation, soil, and climate; differentiated stages of vegetation development; and anthropic activities [5, 6].

In a transition zone between more 'natural' ecosystems, the higher spatial heterogeneity, owing to the presence of representatives of the two communities, tends to lead to greater ecological diversity than its core areas [7]. This situation generates a pattern consistent with the hypothesis of intermediate disturbance, although the high diversity present in ecotones is not necessarily proof of this hypothesis [8, 9]. Thus, spatial heterogeneity in ecotones tends to be reflected in greater degrees of complexity and resilience in these areas [7].

Landscape fragmentation processes caused by anthropic actions tend to create more abrupt transition zones, which are responsible for negative impacts related to edge effects [10, 11]. Therefore, these are areas of high environmental stress and represent possible threats to the resilience of the landscape, as they can affect the self-organisation of the system. Anthropogenic activities related to the removal of natural vegetation, as well as extensive and intensive land use conversion [11–14], are responsible for the increase in landscape fragmentation and expansion of their transition zones. Habitat fragmentation is a major threat to landscape connectivity [15] and contributes a large human footprint in the Anthropocene, especially in tropical forests [16].

Four fundamental aspects can be highlighted regarding transition areas and increased fragmentation of landscapes: (i) structurally, the fragments often constitute important areas of biodiversity and endemism, especially in tropical regions [17–19]; (ii) from a dynamic environmental viewpoint, levels of stability, resistance, and resilience can be estimated via an integrated study of fragments and transition zones that correspond to the responses produced

by the ecosystem to anthropic or natural disasters [20–22]; (iii) processes related to the vegetation development phase, chemical weathering, and erosion are usually rapid in tropical regions, making these environments extremely vulnerable to rapid modifications; (iv) fragmentation processes are indicators of transformation on a global scale caused by alterations in processes such as carbon fixation, nutrient cycle, atmospheric gas permutation (e.g. N, $CO_2$, and O), albedo levels, and evapotranspiration [22–25].

Thus, landscape transition zones, formed due to anthropic fragmentation, represent the interconnections between the physical-natural subsystem (nature) and the socioeconomic subsystem (society) present in this type of complex environmental system. Similar to other complex adaptive systems, the organisation and dynamics of a landscape are the result of non-linear processes that are self-organised in networks, far from thermodynamic equilibrium, and are regulated by cybernetic feedback processes. These characteristics are evidenced in landscape patterns, as there is a direct relationship between patterns and processes [26].

From the perspective of the complexity paradigm, there are several methods to quantify the complexity of patterns (in the sense of spatial heterogeneity) of environmental systems and their elements, as well as determine how they are affected by disturbances, such as processes that cause fragmentation of the landscape. However, as highlighted by Anand et al. [27], although ecologists have historically appropriated the information-theoretic definitions from paradigm of complexity to measure diversity (as is the case with Shannon entropy), there are still few people who have sought to quantify complexity based on other information and coding theoretical definitions proposed by this paradigm. Shannon's entropy of information [28] is a way of measuring the amount of information in a system associated with its possible distribution of probabilities (diversity of information) and how they effectively present themselves in this system. As mentioned by Connor et al. [29], in Shannon entropy "uncertainty is maximized, and information is minimized, when the probability of the observed state of a system may be drawn from a uniform distribution of possibilities (one in which any state of the system is equally probable)." The reason to use information entropy to assess the complexity of systems is given by Newman et al. [30] when they point out that "more complex systems require more information to describe any given state of that system" and, therefore, "complexity and [Shannon] information theory are fundamentally linked".

Parrott [31] presents some of these information entropy-inspired measures to assess the complexity for both temporal and spatial patterns of complex environmental systems and argues that, if appropriate measures are developed and validated, complexity can become a key ecological indicator. Others researches also apply information entropy to evaluate the complexity of a landscape, in addition to its units and transition zones [32–34].

The concept of information entropy is especially interesting when studying landscapes from images of remote sensors, such as satellites and unmanned aerial vehicles (UAVs). The variability in pixel values in a remote sensor image represents the diversity of information present in a landscape and its units and can serve, for example, to estimate the change in the amount of information in the system caused by the fragmentation.

The potential and challenges of using remote sensing while studying landscape fragmentation and loss in resilience, owing to human activities, are well exemplified in areas where the Cerrado occurs. Considered as a biodiversity hotspot, the Cerrado is a vegetation mosaic comprising different phytophysiognomies, from more open ones to forest formations [35]. Throughout Brazil's history, especially in the last five decades, the Cerrado has suffered considerable changes, owing to anthropic actions that have provoked a significant reduction in its original coverage area and to the intense fragmentation of remnants of this vegetation [36]. Thus, as highlighted by Mattos et al. [37], the organisation and dynamics of the Cerrado

characterise it as a complex environmental system whose phytophysiognomies alternate in time and space, owing to natural and anthropic factors and processes.

In this study, we demonstrated how landscape metrics based on information entropy can be applied to assess the complexity (in the sense of spatial heterogeneity) of patches patterns, and their transition zones, present in a Cerrado conservation area and its surroundings, located in south-eastern Brazil (state of São Paulo). This analysis aims to show how such measures can be used to evaluate and indicate how changes in land use and fragmentation caused by such changes affect the complexity of the landscape, and consequently, to indicate how measures based on entropy information can be used as indicators of its integrity and resilience.

## Material and methods

### Study area

The Itirapina Ecological and Experimental Parks (located at coordinates 22˚11'S, 47˚51'W and 22˚15'S, 47˚45'W, respectively), in addition to the surrounding areas including the Lobo dam and the Botelho private reserve (at the boundaries of the towns of Brotas, Itirapina, and São Carlos in the State of São Paulo, Brazil), was selected as the area of study (Fig 1).

This area was chosen because it represents a very heterogeneous landscape with a mosaic of Cerrado phytophysiognomies and intense anthropic activity. The original vegetation consisted largely of savanna and a semi-deciduous seasonal forest with less spatial representation. Owing to the fragmentation caused by intense land use, these two vegetation types have currently been reduced to 16,313 remaining fragments in the State of Sao Paulo [38].

The study area is part of the Parana sedimentary basin, which is a slightly undulating relief, is poorly desiccated, and has low drainage density and declivity [39]. The predominant soils in the region are entisols *(essentially comprising quartz sands), oxisols (dominant in the northern sector; yielded by the decomposition of basaltic rocks; predominant clayey texture), and histosols (organic character; present in low and poorly drained locations). The pedogenical group is mainly the result of weathering of sandstones of the Botucatu and Piramboia Groups [40].

The precipitation patterns observed in Itirapina are similar to those in other Cerrado areas, with seasonal distribution characterised by a rainy season (a wet summer) and a dry winter season (from May to September). The mean annual precipitation was within the range of values obtained by Nimer and Brandão [41] for the Cerrado domain.

To study the Itirapina Ecological and Experimental Parks and their surrounding areas, we used level 5 remote sensing images of the study area, obtained by the MUX sensor of CBRS-4 satellite from 28 July 2018 (orbit point: 156/125). As noted by Martins et al. [42], the MUX sensor has a spatial resolution of 20 m, with four bands (numbered 5, 6, 7, and 8) that correspond to blue, green, red, and near infra-red bands of the electromagnetic spectrum. A MUX level 5 image uses an atmospheric correction algorithm (Coupled Moderate Products for Atmospheric Correction, CMPAC) that was implemented for atmospheric correction of the CBERS MUX level 4 images [42].

## Methods

### Application of complexity measures based on information entropy of remote sensing images

In remote sensing, a digital number (DN) is the value that a pixel of an image has in a particular band, as a function of its intensity of radiation (reflectance or absorbance values after radiometric/atmospheric calibration procedures), for that range of the electromagnetic spectrum

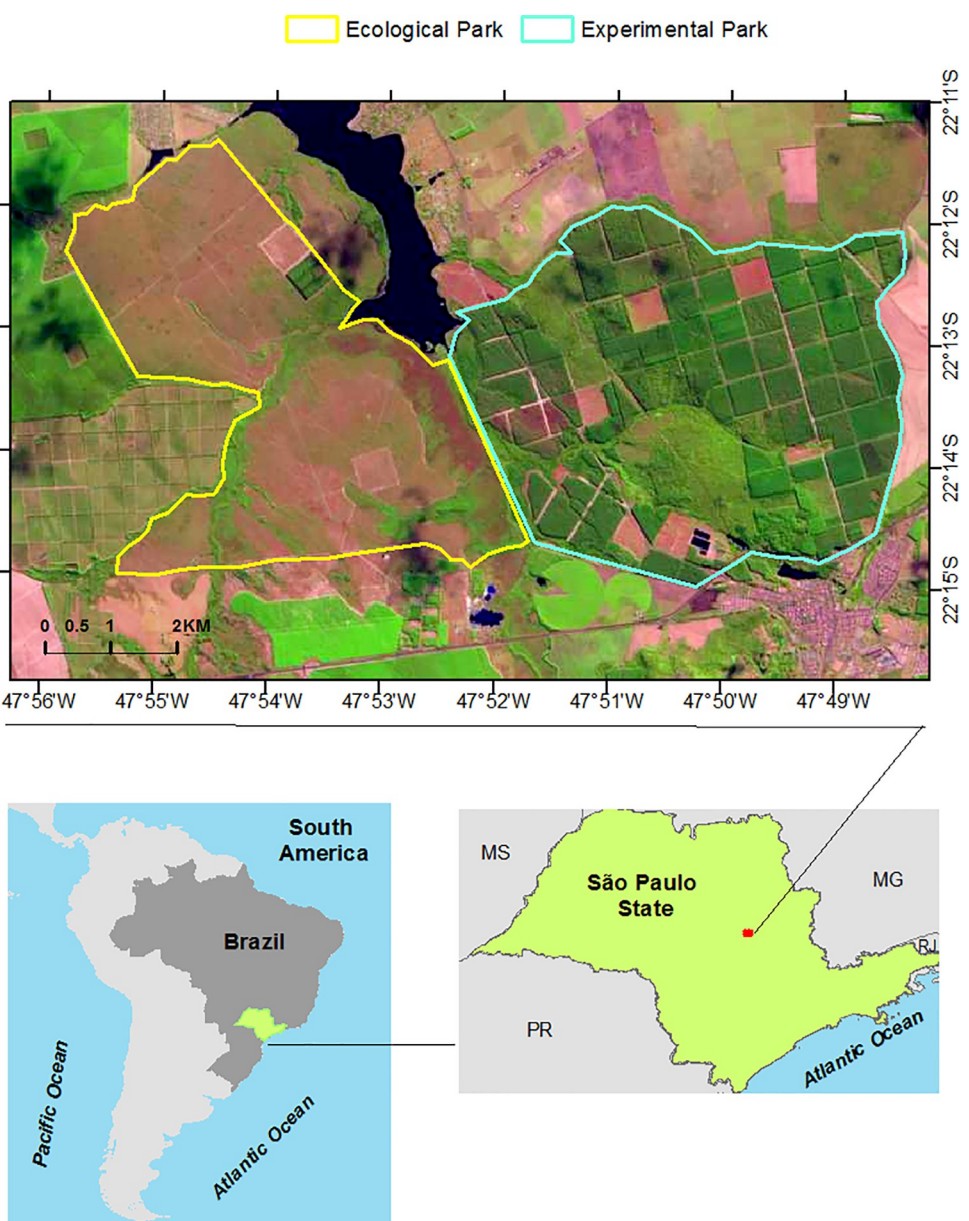

**Fig 1. Location of Itirapina's Ecological and Experimental Parks, São Paulo-Brazil.** (Sentinel-2 (ESA) image courtesy of the U.S. Geological Survey).

[43, 44]. Making an analogy to the Shannon's schematic diagram of a general communication system [28], the DNs (signal) in a remotely sensed image (transmitter) can be used to measure the information contained in a landscape target (source of information) and determine how the intensity of the target surface reflectance (message) can be associated with the complexity of landscape patterns (Fig 2). Moreover, each DN set comprises matrices from data targets, covering different spectral regions (bands of multispectral sensors), which measure the spectral behaviour of targets, compared to electromagnetic radiation, thereby providing a robust set of information from the landscape.

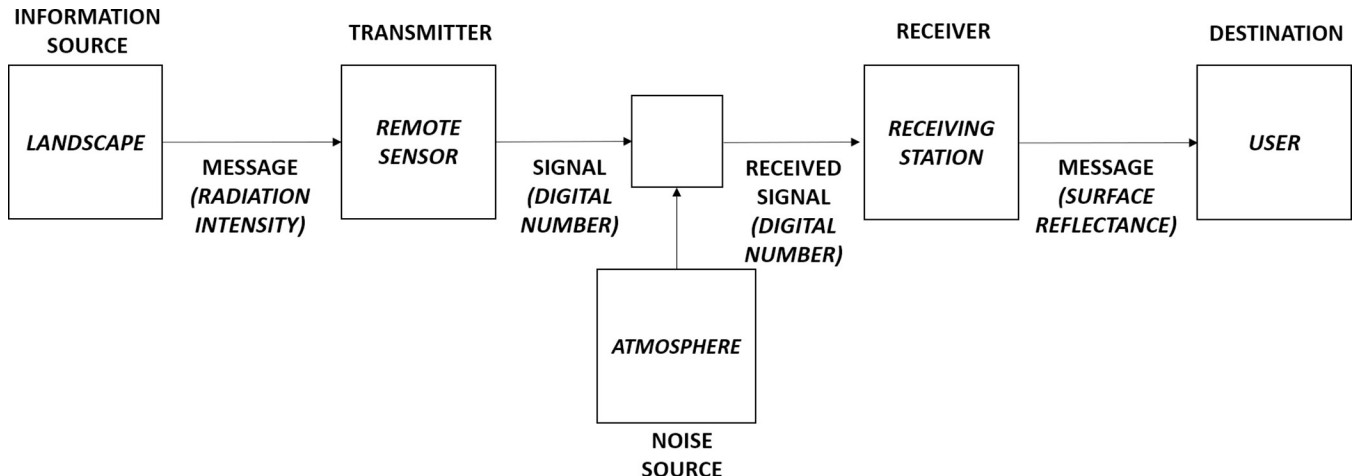

**Fig 2. Shannon's schematic diagram of a general communication system, adapted for information flow in remote sensing (satellite).** (Modified by the authors from Shannon [28]).

Therefore, the number and variability of DNs present in an image or in a region of interest (ROI) of this image can be used to calculate the information entropy, as well as the other measures based on it also used here. Shiner et al. [45] classified complexity measures based on information entropy into three broad categories, according to pattern organisation: I) measures that associate higher values of complexity to disordered patterns; II) measures that assign higher values of complexity to ordered patterns; III) measures represented by a convex function of disorder. However, as highlighted by [46], there is no real complexity in situations that present zero or maximum entropy, and measures belonging to categories I and II are not appropriate for evaluating pattern complexity.

This is especially true for landscape studies, as increased spatial complexity is expected in patterns of intermediate heterogeneity, situated between disordered and ordered patterns [47]. Category 3 measures are those capable of capturing this 'fingerprint' of landscape complexity. In contrast, the use of measures in categories I and II can be useful to assess where a patch of the landscape that does not present intermediate heterogeneity is located in the gradient that has ordered patterns on one extreme and disordered patterns on the other, because in a convex function, the same value appears twice (except for the maximum value, located at the peak of the curve).

When information entropy measure is applied to characterize natural phenomenon, its maximum value corresponds to an equiprobable probability distribution. So, it can be used to evaluate de diversity of the possible patterns. Consequently, if information entropy is used to evaluate a process, it is maximized for systems presenting thermodynamic equilibrium (disorder).

Assuming that complexity is maximized for systems in the half way of the equilibrium (disorder) and disequilibrium (order) [46], two important papers [45, 48] propose to measure complexity in a quantitative way, combining equilibrium and disequilibrium measures, based on information entropy of probability distributions. These measures, called LMC (Lopez-Ruiz, Mancini and Calbet) [48] and SDL (Shiner, Davison and Landsberg) [45], use information entropy [49] to evaluate equilibrium (disorder). Both measures attributes zero complexity to perfect crystal (total order) and to ideal gas (total disorder) [50], considering that LMC measures disequilibrium (order) by the deviation between the considered distribution and the

uniform one. The SDL measures disequilibrium (order) by the complement of the equilibrium (disorder) term. In the literature, these measures are applied to several practical cases [50, 51].

To evaluate the complexity of patterns in the Itirapina Ecological and Experimental Parks in and the surrounding areas, we used two measures belonging to category 1 ($H_e$ and $H_e/H_{max}$ variability measures) and two belonging to category 3 (López-Ruiz, Mancini, and Calbet (LMC) and Shiner, Davison, and Landsberg (SDL)). As described by Mattos [52], the first step in applying these measures to remotely sensed images is the calculation of the system extension ($N$), which represents the number of possible states in the system. Therefore, $N$ is equal to the total of different DNs that an image or ROI has. The maximum entropy ($H_{max}$) is equivalent to the case where DN values (i.e. states) are equiprobable, which corresponds to a situation in which the DN values have the same probability (Eq 1).

$$H_{max} = N \tag{1}$$

The Boltzmann–Gibbs–Shannon entropy ($H_e$) is calculated considering the probability $p$ of the $i^{th}$ DN value within an entire image or ROI (Eq 2), as follows:

$$H_e = -\sum_{DN \in N} P_{(DN)} \log_2 P_{(DN)} \tag{2}$$

In Fig 3, we illustrate two hypothetical situations of remotely sensed images with the same $N$ and $H_{max}$, but different values of $H_e$, owing to differences in the relative frequency of some DNs.

The $H_e/H_{max}$ variability measure is obtained by dividing the calculated entropy ($H_e$) by the maximum entropy ($H_{max}$), as shown in Eq 3:

$$V = \frac{H_e}{H_{max}} \tag{3}$$

For this measure, 0 and 1 are the lowest and the highest values, respectively, corresponding to the extremes of ordered patterns (zero or values close to zero) and disordered patterns (1 or values slightly lower than 1).

To create a convex function of entropy, Shiner et al. [45] proposed SDL measure combining a disorder term and an order term has been proposed (Eq 4):

$$SDL = (H_e/H_{max})[1 - (H_e/H_{max})] \tag{4}$$

Another convex function of entropy (LMC measure) was formulated by López-Ruiz et al. [48], based on a disequilibrium term ($D$), as shown in Eq 5:

$$D = \sum_{i=1}^{N} \left( P_{(DN)} - \frac{1}{N} \right)^2 \tag{5}$$

The LMC measure is given by Eq 6:

$$LMC = \frac{H_e}{H_{max}} \cdot D \tag{6}$$

The minimum values for both SDL and LMC are zero, whereas the maximum values for SDL and LMC are 0.25 and 0.15, respectively.

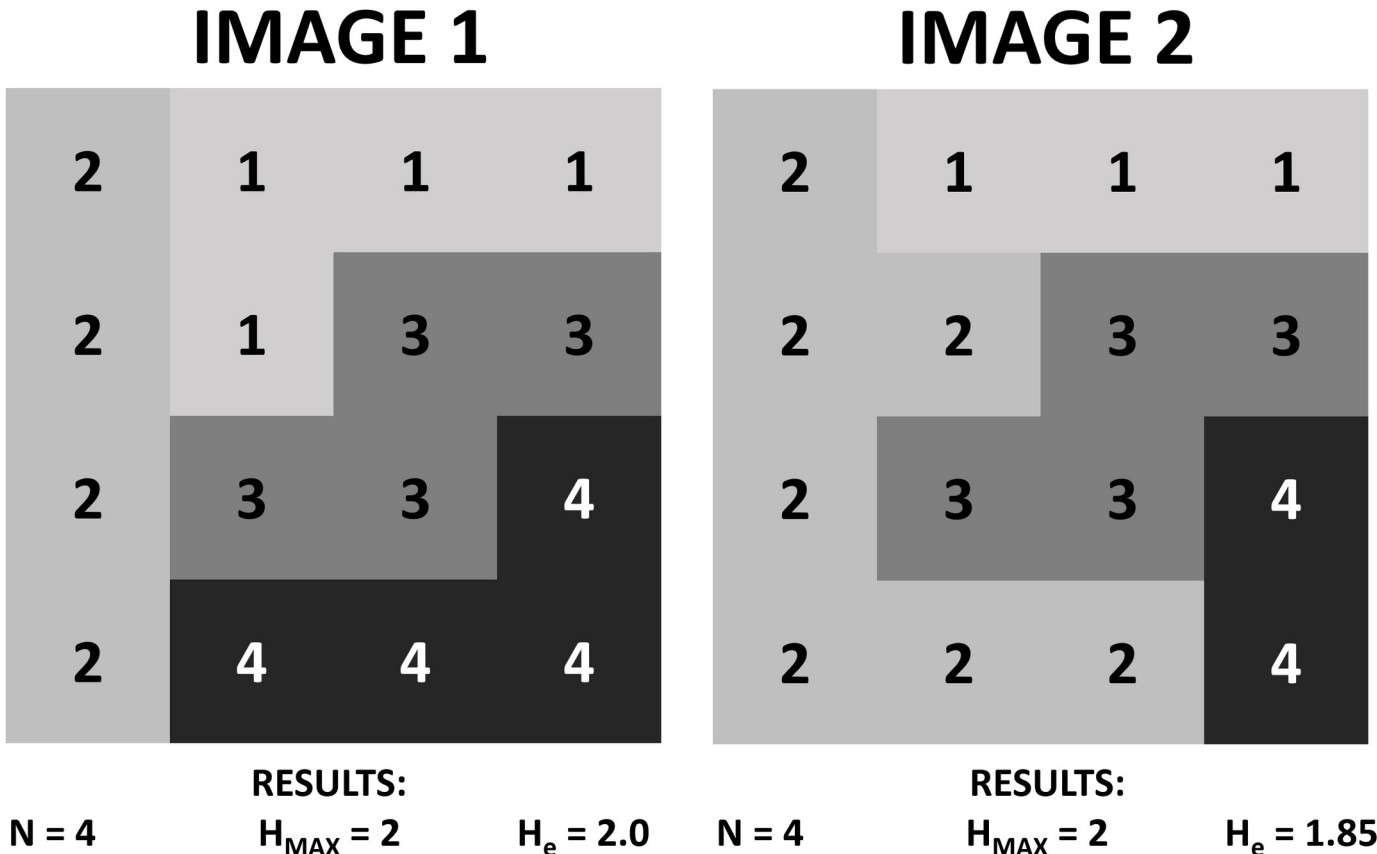

**Fig 3. Maximum entropy ($H_{max}$) and entropy ($H_e$) results for hypothetical images with the same system extension (N), but different frequencies for some digital numbers (DNs).**

### *CompPlex HeROI* and *CompPlex Janus* toolboxes for evaluating complexity of landscape patterns and detecting transition areas in the landscape

*CompPlex HeROI* and *CompPlex Janus* were created to automate the calculations of complexity measures based on information entropy. The term 'CompPlex' aims to integrate the words 'complexity' and 'computation', while the term 'HeROI' is a combination of the acronyms for 'entropy' (He) and 'region of interest' (ROI). In addition, the Portuguese word 'herói' means hero. 'Janus' is an allusion to the god of Roman mythology with two faces: one facing the past and one facing the future. In Portuguese, the term corresponding to window is 'janela', which has its root in the word 'Janus'. '*CompPlex Janus*' is a tribute to Prof. Dr. Sérgio Mascarenhas de Oliveira, the Brazilian scientist who was an enthusiast of Engineering of Complexity and remains a scientific benchmark even after his recent demise [53]. Both scripts were developed in the Python language, to be executed as plugins in the open-source geographic information system QGIS. The toolboxes use GDAL, NUMPY, and PANDAS Python libraries and are included in QGIS via plugins created using the Plugin Builder tool.

*CompPlex HeROI* was used to calculate $H_e$, $H_e/H_{max}$, SDL, and LMC entropy measures in the ROIs previously selected in a remote sensing image. Calculations were performed for all bands of the image (in the case of a multiband satellite), and each ROI was identified using an identifier chosen from the table of attributes of the ROI layer information plan.

The tool is used through a dialogue box that is accessed by the toolbar *CompPlex HeROI* in the QGIS tools area (the toolbar is represented by a star icon). This dialogue box has three

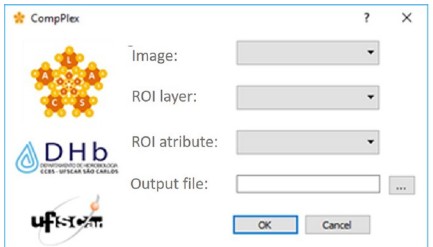
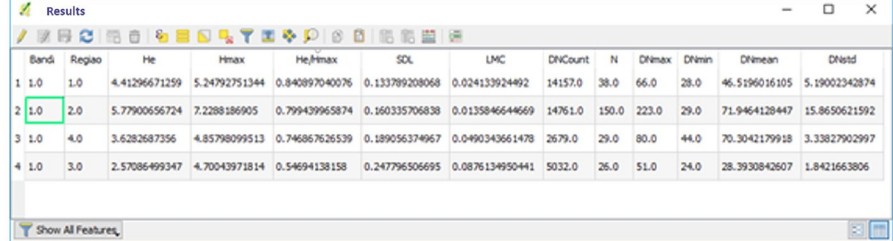

**Fig 4. *CompPlex HeROI* toolbox interface (left) and its results table (right).**

checkboxes: the first two ('Image' and 'ROI layer', respectively) allow the selection of layers present in a QGIS project (Fig 4). The first checkbox has a filter for image-type layers, and the second one has a filter for polygon vector layers. The third checkbox is linked to the second one and allows the selection of one of the fields from the attribute table of the layer selected in the second checkbox. A button opens a saving dialogue box where the directory and file name are selected, with a filter for text files of the CSV type. After selecting the parameters, the tool is executed by clicking on the 'OK' button, following which the calculations are performed and saved in the selected file and a table with the name 'Results' is added to QGIS project, being visible in the layers panel; this table can be opened using the 'Open Attribute Table' command (Fig 4). All algorithm details and support material are accessible at https://github.com/lascaufscar.

The algorithm has two main functions. The first selects the pixels that overlap a feature (polygon corresponding to an ROI) and stores its values in a matrix, and then calculates the values of the following descriptive statistics for the set of selected pixels: count, minimum and maximum values, arithmetic mean, and standard deviation. The second uses the output matrix of the first function to calculate N, $H_e$, $H_{max}$, $H_e/H_{max}$, SDL, and LMC) for that feature.

To perform calculations for all bands and all features, two linked loops are used: the first for the number of bands in the image and the second for the number of features of the ROI layer. The results of the calculations for each feature are stored in a table-like data structure from the Python Pandas library, and at the end of the loop, this table is converted and saved as a CSV text file.

In the case of *CompPlex Janus*, the script for calculating the metrics of complexity for an whole image and the result is a new image with the values of the applicable metric, that is, a map of landscape complexity. The algorithm works similar to a traditional filtering algorithm. First, the user selects the size of a movable window to perform the calculations using a convolution process (Fig 5). Convolution plays the role of filtering to extract information of interest, from the image to which they are applicable, to the function of the defined metric. Moreover, the use of this filter is performed via matrices called masks or kernels. During the application of convolution in an image, the kernel will move along the image as a movable window and will select the DN values to which the function of the determined metric is applied, and the result of this calculation forms a new image, with its value occupying the central position of the kernel (Fig 6).

## Results and discussion

### Results from *CompPlex HeROI* for different land uses and transition areas

The complexity of the patterns of patches in the Itirapina Ecological and Experimental Parks and different land uses of surrounding areas were evaluated by applying *CompPlex HeROI* for ROIs of several land uses and land cover (Fig 7), which are designated as follows:

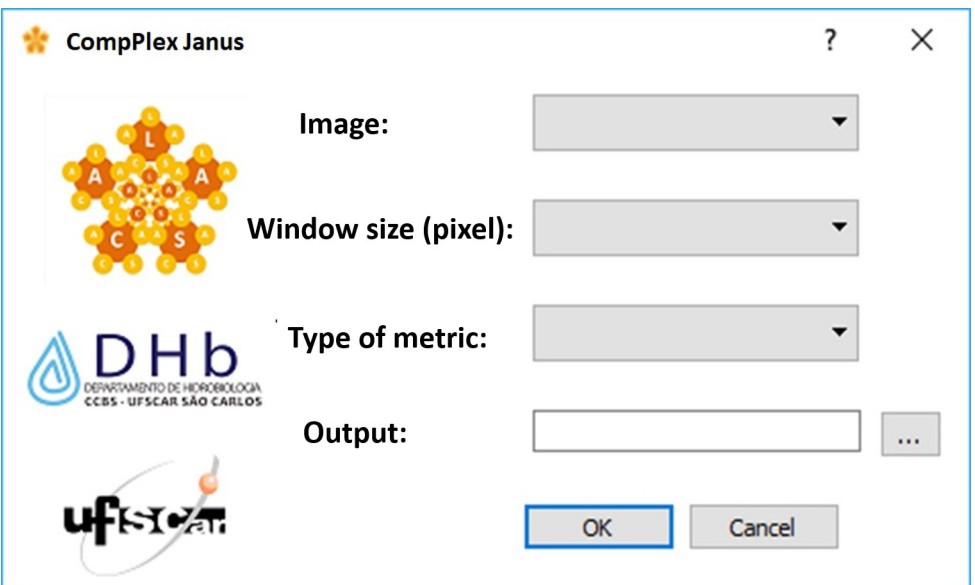

**Fig 5. *CompPlex Janus* toolbox interface.**

- grassland: a physiognomy covered only by the herbaceous extract, locally named as 'campo limpo' (clean field);

- savannah type 1: characterised by a grassland with sparse shrubs (called 'campo sujo', a Portuguese term for dirty field);

- savannah type 2: woodland with closed shrubs and sparse trees (known as 'cerrado stricto sensu');

- native forest type 1: corresponding to a Cerrado forest (Cerradão, i.e. 'big cerrado');

- native forest type 2: 'gallery forest', a woodland located on a river bank;

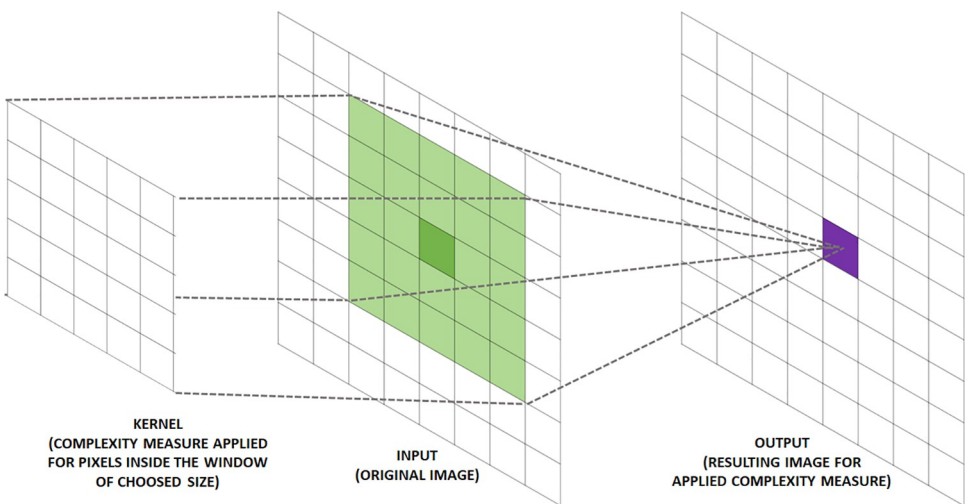

**Fig 6. Operation of *CompPlex Janus* script.**

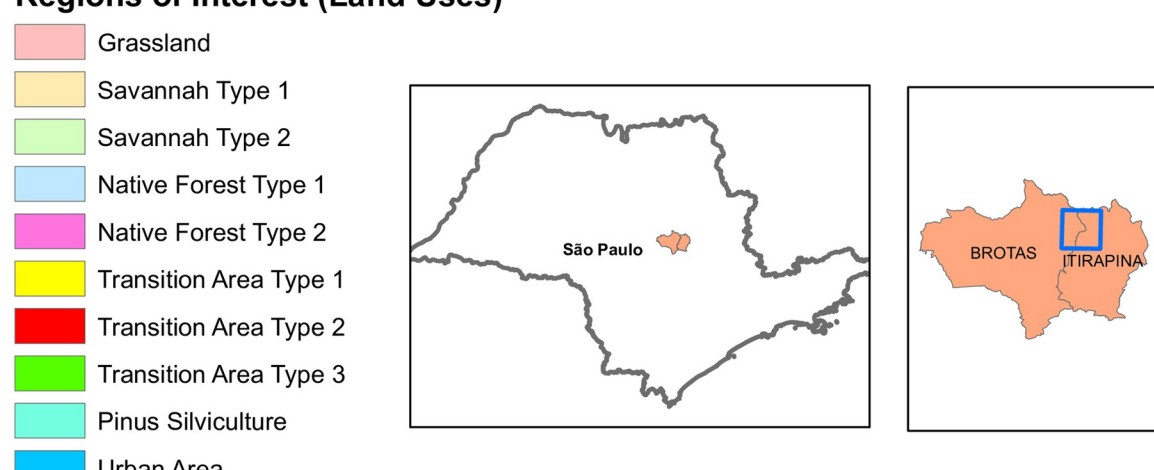

## Legend

### Regions of Interest (Land Uses)

- Grassland
- Savannah Type 1
- Savannah Type 2
- Native Forest Type 1
- Native Forest Type 2
- Transition Area Type 1
- Transition Area Type 2
- Transition Area Type 3
- Pinus Silviculture
- Urban Area

**Fig 7. Regions of interest (ROIs) selected from the Itirapina Ecological and Experimental Parks and their surroundings (State of Sao Paulo, Brazil).** (CBERS-4 image courtesy of the Brazilian National Institute for Space Research).

- *Pinus* silviculture: corresponding to an unmanaged area of *Pinus* spp. with different levels of development;

- urban area: encompassing a part of Itirapina town;

- transition area type 1: an area that includes three different land uses, i.e. *Pinus* spp., grassland ('clean field'), and bare soil;

- transition area type 2: this type is composed of woodland, wetland, and bare soil;

- transition area type 3: corresponding to a mixture of *Pinus* spp., cerrado *stricto sensu*, and bare soil.

Results of complexity measures based on information entropy obtained for these areas are shown in Tables 1–3. Comparing the values of the $H_e/H_{max}$ variability measure for different land uses and remote sensor bands, it can be noted that although values for bands 5, 6, and 7 are obviously not equal, land uses with higher values (more reddish tones in Table 1) are, in general, the same for these three bands, as well as in relation to land uses with lower values (more greenish tones). It can also be observed that for these three bands, the lowest values are associated with vegetation with a greater presence of the arboreal stratum, which provides less variability than uses where this stratum is absent or is not very dense. Similar results were found in another region by Mattos et al. [52], who compared, among other uses, areas with forest formation, with agriculture and with pasture, and the former had lower values for the He/ Hmax measure than the other two types of use. Thus, the use of this measure allows detecting the loss of complexity caused by the conversion of natural areas into agro-pastoral areas.

The results of this algorithm are extremely logical, according to the spectral band arrangement (Table 4) typically associated with soil compounds, as a dataset clustering in an opposite

**Table 1. Results of *CompPlex HeROI* for variability measure ($H_e/H_{max}$) applied to different landscape uses from the Itirapina Ecological and Experimental Parks and their surroundings.** (State of Sao Paulo, Brazil).

| | Measure | | | | | $H_e/H_{max}$ | | |
|---|---|---|---|---|---|---|---|---|
| | Band** | 5 | 6 | 7 | 8 | Stats | | |
| | | | | | | *Mean* | *Standard deviation* | *Variation coefficient* |
| Land use* | Grassland | 7,61E-01 | 7,96E-01 | 8,34E-01 | 8,09E-01 | 8,00E-01 | 3,05E-02 | 3,82E-02 |
| | Savannah type 1 | 7,80E-01 | 7,93E-01 | 8,84E-01 | 8,64E-01 | 8,30E-01 | 5,14E-02 | 6,19E-02 |
| | Savannah type 2 | 6,52E-01 | 6,97E-01 | 7,89E-01 | 6,92E-01 | 7,07E-01 | 5,78E-02 | 8,17E-02 |
| | Native forest type 1 | 6,75E-01 | 5,50E-01 | 6,90E-01 | 7,93E-01 | 6,77E-01 | 9,95E-02 | 1,47E-01 |
| | Native forest type 2 | 6,69E-01 | 6,05E-01 | 6,00E-01 | 8,88E-01 | 6,91E-01 | 1,35E-01 | 1,96E-01 |
| | Transition area type 1 | 9,36E-01 | 9,21E-01 | 9,30E-01 | 8,92E-01 | 9,20E-01 | 1,94E-02 | 2,11E-02 |
| | Transition area type 2 | 7,09E-01 | 7,65E-01 | 8,96E-01 | 8,19E-01 | 7,97E-01 | 7,99E-02 | 1,00E-01 |
| | Transition area type 3 | 7,51E-01 | 7,72E-01 | 7,68E-01 | 9,19E-01 | 8,02E-01 | 7,84E-02 | 9,78E-02 |
| | *Pinus* silviculture | 6,17E-01 | 5,54E-01 | 6,56E-01 | 8,75E-01 | 6,75E-01 | 1,40E-01 | 2,07E-01 |
| | Urban area | 7,45E-01 | 8,27E-01 | 8,71E-01 | 7,52E-01 | 7,99E-01 | 6,08E-02 | 7,61E-02 |
| Stats | *Mean* | 7,29E-01 | 7,28E-01 | 7,92E-01 | 8,30E-01 | | | |
| | *Standard deviation* | 8,98E-02 | 1,23E-01 | 1,12E-01 | 7,10E-02 | | | |
| | *Variation coefficient* | 1,23E-01 | 1,70E-01 | 1,41E-01 | 8,55E-02 | | | |

* Grassland = 'clean field'; savannah type 1 = 'dirty field'; savannah type 2 = 'cerrado *stricto sensu*'; native forest type 1 = 'big cerrado'; native forest type 2 = gallery forest; transition area type 1 = *Pinus* silviculture + 'clean field' + bare soil; transition area type 2 = woodland + bare soil + wetland; transition area type 3 = *Pinus* silviculture + cerrado *stricto sensu* + bare soil.

** Bands 5, 6, 7, and 8 correspond to the blue, green, red, and near infrared bands, respectively, of electromagnetic spectrum.

**Table 2. Results of *CompPlex HeROI* for the SDL measure applied to different landscape uses from the Itirapina Ecological and Experimental Parks and their surroundings (State of Sao Paulo, Brazil).**

| | Measure | SDL | | | | | | |
|---|---|---|---|---|---|---|---|---|
| | Band** | 5 | 6 | 7 | 8 | *Stats* | | |
| | | | | | | *Mean* | *Standard deviation* | *Variation coefficient* |
| Land use* | Grassland | 1,82E-01 | 1,62E-01 | 1,38E-01 | 1,54E-01 | 1,59E-01 | 1,81E-02 | 1,14E-01 |
| | Savannah type 1 | 1,72E-01 | 1,64E-01 | 1,03E-01 | 1,17E-01 | 1,39E-01 | 3,41E-02 | 2,45E-01 |
| | Savannah type 2 | 2,27E-01 | 2,11E-01 | 1,67E-01 | 2,13E-01 | 2,04E-01 | 2,62E-02 | 1,28E-01 |
| | Native forest type 1 | 2,19E-01 | 2,47E-01 | 2,14E-01 | 1,64E-01 | 2,11E-01 | 3,47E-02 | 1,64E-01 |
| | Native forest type 2 | 2,21E-01 | 2,39E-01 | 2,40E-01 | 9,95E-02 | 2,00E-01 | 6,75E-02 | 3,38E-01 |
| | Transition area type 1 | 5,99E-02 | 7,31E-02 | 6,54E-02 | 9,63E-02 | 7,36E-02 | 1,60E-02 | 2,18E-01 |
| | Transition area type 2 | 2,06E-01 | 1,80E-01 | 9,31E-02 | 1,48E-01 | 1,57E-01 | 4,87E-02 | 3,11E-01 |
| | Transition area type 3 | 1,87E-01 | 1,76E-01 | 1,78E-01 | 7,43E-02 | 1,54E-01 | 5,33E-02 | 3,46E-01 |
| | *Pinus* silviculture | 2,36E-01 | 2,47E-01 | 2,26E-01 | 1,09E-01 | 2,05E-01 | 6,41E-02 | 3,13E-01 |
| | Urban área | 1,90E-01 | 1,43E-01 | 1,12E-01 | 1,86E-01 | 1,58E-01 | 3,71E-02 | 2,35E-01 |
| *Stats* | *Mean* | 1,90E-01 | 1,84E-01 | 1,54E-01 | 1,36E-01 | | | |
| | *Standard deviation* | 5,05E-02 | 5,45E-02 | 6,06E-02 | 4,41E-02 | | | |
| | *Variation coefficient* | 2,66E-01 | 2,95E-01 | 3,95E-01 | 3,24E-01 | | | |

* Grassland = 'clean field'; savannah type 1 = 'dirty field'; savannah type 2 = 'cerrado *stricto sensu*'; native forest type 1 = 'big cerrado'; native forest type 2 = gallery forest; transition area type 1 = *Pinus* silviculture + 'clean field' + exposed soil; transition area type 2 = woodland + bare soil + wetland; transition area type 3 = *Pinus* silviculture + cerrado *stricto sensu* + exposed soil.

** Bands 5, 6, 7 and 8 correspond, respectively, to blue, green, red and near infrared bands of electromagnetic spectrum.

**Table 3. Results of *CompPlex HeROI* for LMC measure applied to different landscape uses from the Itirapina Ecological and Experimental Parks and their surroundings (State of Sao Paulo, Brazil).**

| | Measure | LMC | | | | | | |
|---|---|---|---|---|---|---|---|---|
| | Band** | 5 | 6 | 7 | 8 | *Stats* | | |
| | | | | | | *Mean* | *Standard deviation* | *Variation coefficient* |
| Land use* | Grassland | 8,52E-02 | 6,25E-02 | 4,88E-02 | 5,45E-02 | 6,28E-02 | 1,60E-02 | 2,54E-01 |
| | Savannah type 1 | 9,25E-02 | 8,93E-02 | 4,07E-02 | 4,68E-02 | 6,73E-02 | 2,73E-02 | 4,06E-01 |
| | Savannah type 2 | 1,48E-01 | 1,21E-01 | 7,74E-02 | 1,21E-01 | 1,17E-01 | 2,93E-02 | 2,51E-01 |
| | Native forest type 1 | 1,38E-01 | 1,64E-01 | 1,28E-01 | 7,49E-02 | 1,26E-01 | 3,75E-02 | 2,97E-01 |
| | Native forest type 2 | 1,50E-01 | 1,47E-01 | 1,46E-01 | 3,23E-02 | 1,19E-01 | 5,77E-02 | 4,86E-01 |
| | Transition area type 1 | 2,85E-02 | 2,99E-02 | 1,70E-02 | 3,91E-02 | 2,86E-02 | 9,02E-03 | 3,15E-01 |
| | Transition area type 2 | 9,42E-02 | 8,13E-02 | 2,87E-02 | 4,55E-02 | 6,24E-02 | 3,05E-02 | 4,88E-01 |
| | Transition area type 3 | 9,70E-02 | 7,87E-02 | 7,48E-02 | 2,10E-02 | 6,78E-02 | 3,27E-02 | 4,82E-01 |
| | *Pinus* silviculture | 1,68E-01 | 1,78E-01 | 1,34E-01 | 3,96E-02 | 1,30E-01 | 6,32E-02 | 4,86E-01 |
| | Urban área | 6,41E-02 | 4,02E-02 | 2,46E-02 | 5,03E-02 | 4,48E-02 | 1,67E-02 | 3,72E-01 |
| *Stats* | *Mean* | 1,07E-01 | 9,93E-02 | 7,20E-02 | 5,26E-02 | | | |
| | *Standard deviation* | 4,37E-02 | 5,14E-02 | 4,84E-02 | 2,81E-02 | | | |
| | *Variation coefficient* | 4,10E-01 | 5,18E-01 | 6,73E-01 | 5,35E-01 | | | |

* Grassland = 'clean field'; savannah type 1 = 'dirty field'; savannah type 2 = 'cerrado *stricto sensu*'; native forest type 1 = 'big cerrado'; native forest type 2 = gallery forest; transition area type 1 = *Pinus* silviculture + 'clean field' + exposed soil; transition area type 2 = woodland + exposed soil + wetland; transition area type 3 = *Pinus* silviculture + cerrado *stricto sensu* + bare soil.

** Bands 5, 6, 7, and 8 correspond to the blue, green, red, and near infra-red bands, respectively, of electromagnetic spectrum.

way to that of vegetational targets values, as shown in Tables 1 and 2. It is possible to see this dataset cluster trend comparing the results of bands 5, 6, and 7 with those of band 8, resulting in an inversion in the relative values of some land uses, as some uses that have higher relative values for bands 5, 6, and 7 present lower relative values for band 8 and vice versa. Of note, band 8 (near infrared: 0.76–0.90 μm) mainly covers the spectral range associated with canopy structure (above ground biomass) (Table 4). This can be seen in the cases of vegetation types being located at the extremes of a gradient, between more open areas (grassland) and more closed areas (native forest type 2 and *Pinus* silviculture). Thus, as band 8 corresponds to near infrared, it can identify variability in plant biomass within the same patch (Table 4). The use of the near infrared band of remote sensors is well established in the literature, especially for the assessment of visible light absorption by the vegetation canopy applying vegetation indices [54]. Moraes et al. [55] used the He/Hmax and SDL measurements in an area with different agricultural management to image the Rapideye sensor in different bands and found that the near infrared had higher values for the He/Hmax measurement in relation to the other bands analyzed (red and red edge).

Transition areas show different behaviours for the $H_e/H_{max}$ variability measure. The type 1 transition area obtained high values for the $H_e/H_{max}$ variability measure for all bands, whereas for types 2 and 3, the values were comparatively low for bands 5 and 6. However, for bands 7 and 8, there was an inversion in their behaviour: the type 2 transition area had a high value in band 7 and low value in band 8, whereas type 3 showed the opposite behaviour. Additionally, as presented in Table 1, compared to other bands, band 8 had the highest average value for the $H_e/H_{max}$ variability measure, but the lowest values for standard deviation and coefficient of variation, among all four bands. These results indicate that the complexity of the patterns of transition areas are dependent on the combination of land uses present in each region, resulting in greater or lesser variability in the quantity and frequency of pixels, which corresponds directly to spectral band cover. These results are important to analyze the process of landscape fragmentation by anthropic actions and on the Intermediate Disturbance Theory, since the type of land use conversion and its intensity can cause a drastic drop in the degree of complexity of the landscape due to decrease in their resilience and integrity.

SDL and LMC complexity measures showed similar behaviours, both in relation to different land uses and the bands (Tables 2 and 3), In contrast to the results obtained for the $H_e/H_{max}$ variability measure, savannah type 2 ('cerrado *stricto sensu*'), native forest type 1 ('big cerrado'), native forest type 2 ('gallery forest'), and *Pinus* silviculture obtained high values for bands 5, 6, and 7, whereas for band 8, the first two types of land use (savannah type 2 and

**Table 4. Comparison of MUX-CBERS 4 and OLI-Landsat 8 sensor bands and their mapping utilities.**

| CBERS 4 (MUX) | | Landsat 8 (OLI) | | |
|---|---|---|---|---|
| Band* | Wavelength (μm)* | Band* | Wavelength (μm)* | Mapping uses** |
| 5 (Blue) | 0.450–0.520 | 2 (Blue) | 0.452–0.512 | Bathymetric mapping, distinguishing soil from vegetation and deciduous from coniferous vegetation |
| 6 (Green) | 0.520–0.590 | 3 (Green) | 0.533–0.590 | Emphasises peak vegetation, which is useful for assessing plant vigour |
| 7 (Red) | 0.630–0.690 | 4 (Red) | 0.636–0.673 | Discriminates vegetation slopes |
| 8 (Near Infrared) | 0.770–0.890 | 5 (Near Infrared) | 0.851–0.879 | Emphasises biomass content and shorelines |

Sources:

* [42].

** [56].

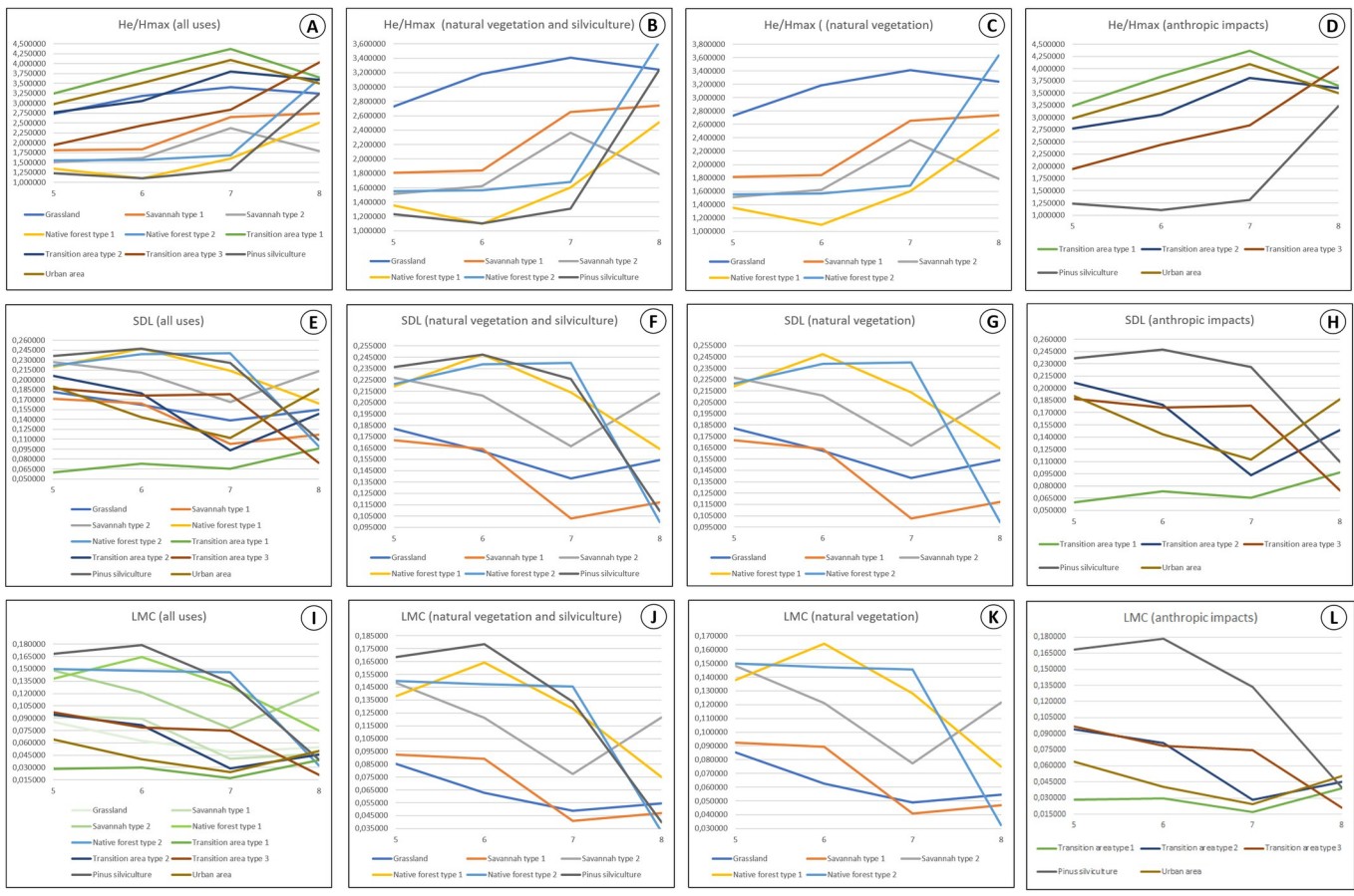

**Fig 8. Complexity signatures for variability, SDL, and LMC measures of different landscape uses from the Itirapina Ecological and Experimental Parks and their surroundings (State of Sao Paulo, Brazil).**

native forest type 1) achieved high values for the measure and the gallery forest (native forest type 2) and *Pinus* silviculture obtained low values. Unlike the results obtained for the $H_e/H_{max}$ variability measure, band 7 had the highest variation coefficient for the SDL and LMC complexity measures.

In contrast, the SDL and LMC complexity measures of the transition were comparatively similar to those of the variability measure, including the differences found between bands 7 and 8 for transition area types 2 and 3 (Tables 2 and 3). Band 3 showed the highest value of variation coefficient, whereas band 4 had the lowest mean and standard deviation values.

The results obtained from *CompPlex HeROI* can also be analysed using the 'complexity signature' of each ROI, which is plotted by placing the sensor bands on the x axis and the measurement values, based on the selected informational entropy, on the y axis. Some examples of complexity signature graphs are shown in Fig 8. Regarding the $H_e/H_{max}$ complexity signatures of all ROIs (Fig 8A), certain patterns related to the behaviour of ROIs are evident. Land uses such as grassland, savannah type 2, urban area, and transition area types 1 and 2 show increases in $H_e/H_{max}$ from band 5 to 7, followed by an inflection to band 8. Moreover, transition area type 3, *Pinus* silviculture, and native forest types 1 and 2 show a gradual increase in $H_e/H_{max}$ from band 6 to 8. When the considered ROIs are only those with vegetation (Fig 8B and 8C), similar patterns are noticeable for native forest types 1 and 2 and *Pinus* silviculture. In addition, it is worth noting the high values of grassland, in relation to those of the other

types of vegetation, in the first three bands, were later surpassed by those of native forest type 2, while approaching the value of *Pinus* silviculture in band 8. Upon analysing the complexity signatures of the ROIs related to anthropic impacts (Fig 8D), two patterns showed a clear differentiation between these uses, the first of which is presented by the urban area and transition area types 1 and 2, and second being presented by *Pinus* silviculture and transition area type 3. These results indicate that it is not appropriate to make generalizations about the role of anthropogenic disturbances on the landscape scale, as each type of human activity and the impacts caused by it can have different results in the complexity of the spatial patterns of this mosaic as a result of possible changes in the processes that form them.

The complexity signatures obtained for the SDL and LMC measures (Fig 8E–8L) showed some similarities in their behaviour, as well as differences from those generated for $H_e/H_{max}$ measure (Fig 8A–8D). However, further subtle differences could be observed when analysing the complexity signatures of some land uses grouped into categories. For ROIs associated with vegetation, the LMC measure was able to better separate some types of vegetation, specifically those having signatures of complexity, in some stretches, which nearly overlapped with or approximated the curves plotted for the SDL measure; this was the case between native forest types 1 and 2, as well as between grassland and savannah type 1 (Fig 8F, 8G, 8J and 8K). In contrast, for land uses related to anthropic impacts (Fig 8H and 8l), the complexity signatures generated by the SDL measure showed fewer overlaps and approximations than those observed for the LMC measure, for which there was at least one case like this comparing each pair of uses.

## Landscape complexity maps generated by *CompPlex Janus*

Complexity maps can also be used to analyse the complexity of landscape metrics [44] and can highlight spatial patterns across several land uses, thereby allowing for full landscape analysis to performed. Figs 9 and 10 show the landscape complexity maps for different measures, wherein window sizes and bands highlight different aspects of the landscape of the Itirapina Ecological and Experimental Parks and their surroundings. For example, landscape complexity maps generated using the $H_e$ measure could mark the limits of the different patches well, both for smaller windows ($3 \times 3$ pixels and $5 \times 5$ pixels; Fig 9A and 9E) and for those having a larger size ($7 \times 7$ pixels and $9 \times 9$ pixels; Fig 9I and 9M). As expected, the smaller window can delimit these edges with a thin line, which will become thicker and more blurred with an increase in the window size, for most cases, and for some cases, the edges might disappear.

However, the complexity maps generated by a $3 \times 3$ pixel window for the $H_e/H_{max}$, SDL, and LMC measures (Fig 9B–9D) do not clearly show the edges of the patches, leaving the maps with 'speckled' dots. However, for larger windows applied to these three measurements, the edges are less blurred, and the heterogeneities for SDL and LMC measures within each patch are more evident than those for the $H_e/H_{max}$ measure.

When comparing maps generated by different bands for the same measurement (Fig 10), regardless of the size of the window, maps generated by bands 5, 6, and 7 show more similarities to each other than those generated by band 8. Vegetation areas, especially those with a more closed canopy (whether they are more 'natural' or planted) had noticeably low values for the $H_e$ and $H_e/H_{max}$ measures and high values for the SDL and LMC complexity measures (Figs 9 and 10), which agreed with the observations noted for areas analysed using the *CompPlex HeROI* script. In *CompPlex Janus* results, this behaviour is better evidenced for the larger windows and for bands 6 and 8 (Fig 10). Notably, for band 8, the generated complexity maps are able to differentiate areas with 'natural' vegetation (Cerrado physiognomies), which have higher values for SDL and LMC measures, from reforestation areas (*Pinus* spp.).

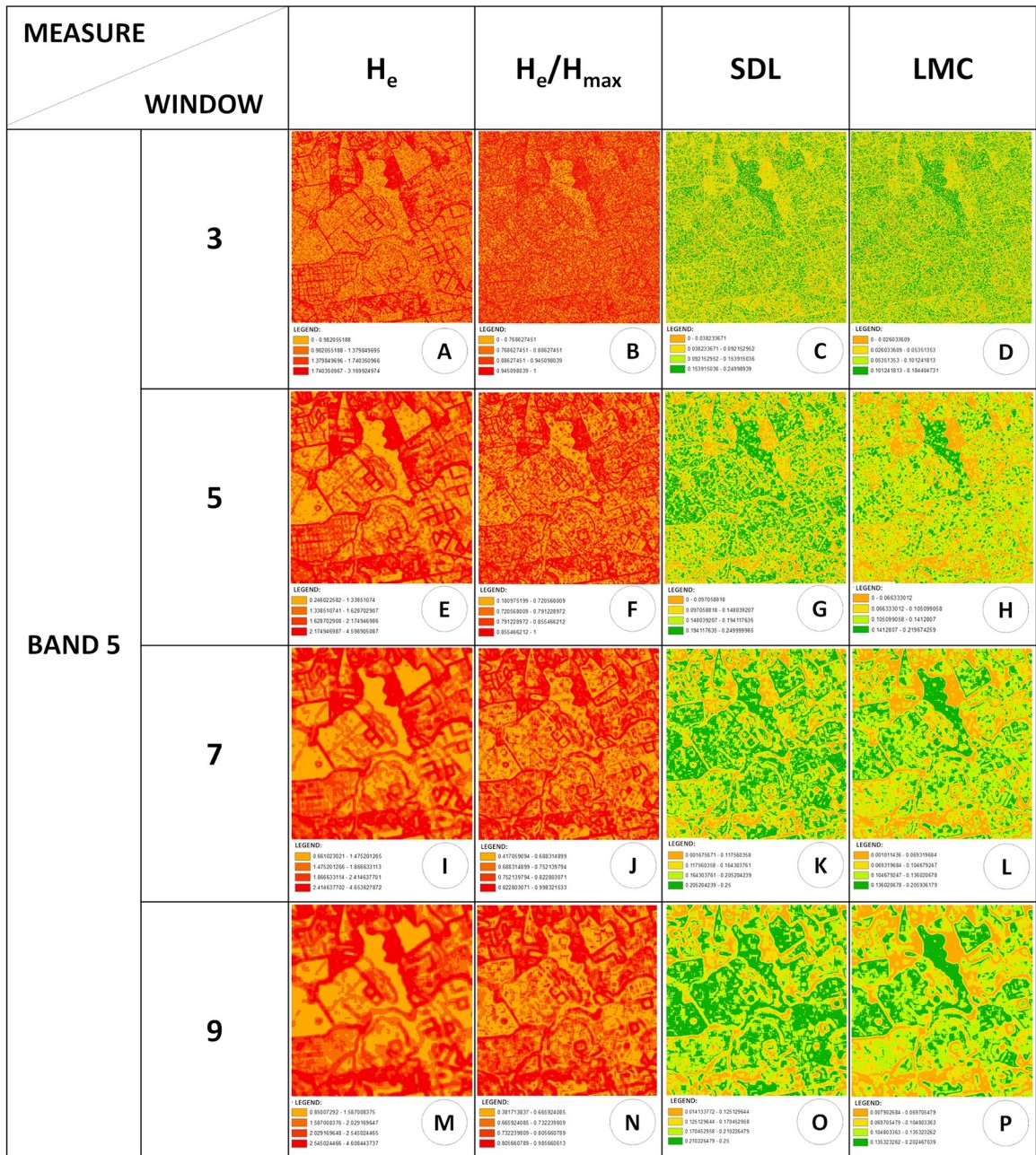

**Fig 9. Landscape complexity maps generated by *CompPlex Janus* for He, He/H_{max}, SDL, and LMC measures applied to different window sizes for band 5, using a CBERS MUX level-4 image of Itirapina Ecological and Experimental Parks and their surroundings (State of Sao Paulo, Brazil).**

Regarding the transition areas, behaviors similar to those obtained here were achieved by Moraes et al. [57] for the complexity maps for the He/Hmax and SDL measurements in a region with conditions similar to those studied here (a conservation unit surrounded by transition areas of different land uses), showing the coherence of the results generated by these measures. This also happened in a study by Mattos et al. [52] for the same area

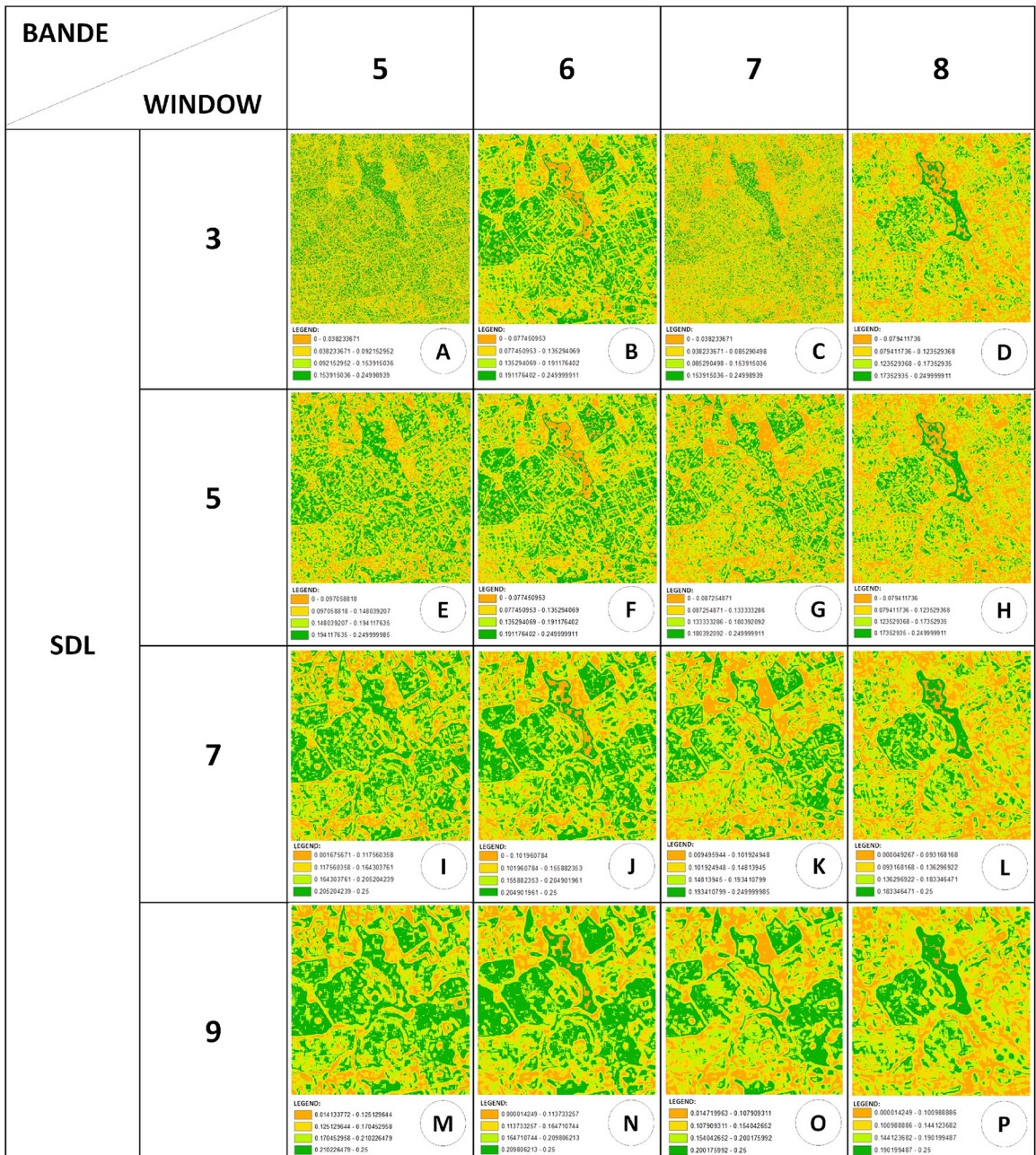

**Fig 10. Landscape complexity maps generated by *CompPlex Janus* for SDL measure applied to different window sizes and bands for a CBERS MUX level-4 image of Itirapina Ecological and Experimental Parks and their surroundings (State of Sao Paulo, Brazil).**

researched by Moraes et al. [57] using the CompPlex Janus, which observed behaviors similar to those reported here in relation to the window size and the CBERS satellite band used, suggesting that both the identification of borders and transition areas and the distinction of vegetation uses are consistent marks of the complexity measures present in CompPlex Janus.

## Conclusions

The Cerrado in the State of São Paulo is currently constituted by remnants in peripheral areas of its original distribution in relation to the core area of this landscape domain (located in the central region of Brazil) and has already suffered from human impacts longer than this core area of the Cerrado, whose 'anthropization' is more recent. Thus, studies such as this, which seek to assess the complexity of the landscape in areas where the Cerrado has already been largely converted to other types of land use, can serve as a reference on the impacts on integrity and resilience that can be caused by the expansion of the area of agriculture in its core area, as well as in the new agricultural frontiers that shelter areas of cerrado (North and Northeast of Brazil). Furthermore, the methodological procedures used here, applying complexity measures based on informational entropy, can be used in other regions of the world where the aim is to assess the impacts of changes in land use.

The results of the current study demonstrated that the complexity algorithms are robust, whereas presented coherence and isonomy among spectral and spatial analysis outputs vs landscape measures/features, allowing: (i) the use of raw images, not requiring complex image pre-processing, (ii) measuring the variability in information levels from landscapes, not only in terms of spatial datasets but also spectrally. The complexity algorithms were able not only to qualify, but to quantify the landscape features, offering different outputs of analytical results, as: tables, graphics and maps, from a single satellite image. Consequently, more spectral/spatial information can reliably be generated for different land uses corresponding to the same complexity landscape measurements. The complexity algorithms can capture the essence of landscape patterns, while preserving the main informational data from each spectral band, thereby enabling the effective use of this mathematical approach for large-scale analysis (for mapping and monitoring land uses) and improving the digital imaging process by reducing the time used by computational processes.

From an ecological viewpoint, the measures, based on information entropy evaluated via the *CompPlex HeROI* and *CompPlex Janus* scripts to assess the complexity of the landscape and its patches, are suitable indicators for quantifying integrity and resilience of these complex environmental systems. So, for policy makers and/or government institutions the algorithm results means a remote sensing + image process method (fast, simple and chip), which provide the auxiliary auditable information able to cover large areas, which is quite important to monitoring, report and verification applications, for instance [58] (MRV, 2018). In this way, the algorithm has a typical remote sensing approach limitation. it is not being a fully analytical approach, reducing substantially but not totally the dependency of a traditional field inventories.

The contrast of behaviors of different land uses, including the typical characteristics of each type of vegetation and each transition area, captured by complexity measures based on informational entropy indicates that these measures adequately reflect the level of complexity (in terms of spatial heterogeneity) of the landscape. As such, such measurements can be used as indicators of the resilience and health of a landscape and its patches. The use of images generated from indices that highlight certain targets in the images of remote sensors, such as those applied to vegetation, can bring an even greater refinement in the differentiation of land uses and by complexity measures based on in informational entropy. Furthermore, the creation of complexity indices by joining the results of a measure in different bands and/or the results of the same band for the different measures used here (for example, one of type) and another of type III can further highlight this contrast between the complexity patterns of different land uses.

Similar to the use in remote sensing of spectral signatures to differentiate different land uses, the complexity signatures can be applied to make this kind of distinction, as well as to assess differences between patches of the same use, as done by Mattos et al. [52]. This application can be useful, for example, for defining priority conservation areas, planning ecological corridors linking patches with greater complexities, and evaluating ecological restoration processes.

According we showed, the complexity algorithm works with fully landscape features. Instead, remote sensing traditional image classification methods, demands different algorithms, image pre-processes and high-end specialists to obtain similar results [59–63]. Nowadays, the use of multispectral sensors datasets has been increased and becomes more accessible, including free multisensors time-series [64–66]. The use of this huge amount of dataset opens totally new possibilities to space-time land use/land cover change monitoring (landscape changes), through the policy makers and remote sensing specialist's applications. For future studies, it may also be interesting to verify possible correlations between complexity measures and shape indices, similarly to what Yang et al. [67, 68] have done by associating the spatial distribution of three-dimensional morphological characteristics of in urban area with land surface temperature. The inclusion of these complexity measures based on informational entropy in image classification procedures may also be promising, applying, for example, the RBF neural network prediction model used by Li et al. [69] to assess the demand for urban-industrial use in an urban agglomeration.

Finally, it should be considered that the choice of ROI sizes in CompPlex HeROI and windows in CompPlex Janus is crucial for evaluating the complexity of spatial patterns of different land uses. Very small ROIs and windows will contain few pixels and therefore less and probably less diversity of information, which can make it difficult to differentiate between different land uses. ROIs and very large windows, on the other hand, can encompass a very large amount and diversity of information, leading to a generalization that can prevent the detection of the individual behavior of each use, its internal variations and its transition areas with other uses. Therefore, it is suggested that future studies on the application of complexity measures based on informational entropy used here investigate optimal sizes of ROIs and windows, as well as the most appropriate bands and band indices to detect different processes of interest and their impacts on complexity of the landscape, such as fragmentation and the loss of its resilience and integrity.

## Author Contributions

**Conceptualization:** Sérgio Henrique Vannucchi Leme de Mattos, Luiz Eduardo Vicente, Andrea Koga Vicente, José Roberto Castilho Piqueira.

**Data curation:** Sérgio Henrique Vannucchi Leme de Mattos, Cláudio Bielenki Júnior.

**Formal analysis:** Sérgio Henrique Vannucchi Leme de Mattos, Andrea Koga Vicente, José Roberto Castilho Piqueira.

**Funding acquisition:** Luiz Eduardo Vicente.

**Investigation:** Sérgio Henrique Vannucchi Leme de Mattos, Luiz Eduardo Vicente.

**Methodology:** Sérgio Henrique Vannucchi Leme de Mattos, Luiz Eduardo Vicente, Cláudio Bielenki Júnior, José Roberto Castilho Piqueira.

**Project administration:** Sérgio Henrique Vannucchi Leme de Mattos.

**Software:** Cláudio Bielenki Júnior.

**Supervision:** Sérgio Henrique Vannucchi Leme de Mattos, José Roberto Castilho Piqueira.

**Validation:** Andrea Koga Vicente, Cláudio Bielenki Júnior.

**Visualization:** Sérgio Henrique Vannucchi Leme de Mattos.

**Writing – original draft:** Sérgio Henrique Vannucchi Leme de Mattos, Luiz Eduardo Vicente, Andrea Koga Vicente, José Roberto Castilho Piqueira.

**Writing – review & editing:** Sérgio Henrique Vannucchi Leme de Mattos, Luiz Eduardo Vicente, Andrea Koga Vicente, José Roberto Castilho Piqueira.

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
