## [Decision Letter · Decision Letter 0]

18 Oct 2021

PONE-D-21-31069LANDSCAPE COMPLEXITY METRICS BASED ON INFORMATION ENTROPYPLOS ONE

Dear Dr. Mattos,

Thank you for submitting your manuscript to PLOS ONE. After careful consideration, we feel that it has merit but does not fully meet PLOS ONE’s publication criteria as it currently stands. Therefore, we invite you to submit a revised version of the manuscript that addresses the points raised during the review process.

We look forward to receiving your revised manuscript.

Kind regards,

Jun Yang

Academic Editor

PLOS ONE

Journal Requirements:

2. In your Methods section, please provide additional location information, including geographic coordinates of your field collection site if available.

3. We note that Figures 1 and 7 in your submission contain [map/satellite] images which may be copyrighted. All PLOS content is published under the Creative Commons Attribution License (CC BY 4.0), which means that the manuscript, images, and Supporting Information files will be freely available online, and any third party is permitted to access, download, copy, distribute, and use these materials in any way, even commercially, with proper attribution. For these reasons, we cannot publish previously copyrighted maps or satellite images created using proprietary data, such as Google software (Google Maps, Street View, and Earth). For more information, see our copyright guidelines: http://journals.plos.org/plosone/s/licenses-and-copyright.

a. You may seek permission from the original copyright holder of Figures 1 and 7  to publish the content specifically under the CC BY 4.0 license.  

Additional Editor Comments:

Reviewer 1

1. The title needs to be more detailed.

2. Please highlight the key points in the abstract.

3. The introduction should further highlight what is the motivation of the paper.

4. Material and Methods: 2.2 should be "Methods", "Application of complexity measures based on information entropy of

remote sensing images" etc. should be 2.2.1.

5. Results and discussion: The results should be better described, discussed and justified using references.

6. Conclusion mostly looks like a summary of the work done and the results obtained. No interpretation of the results in given as well as no recommendation for the government and policy makers as to how the results could be used. Also, there should be some content in the conclusion regarding the limitations of the current research and future work possibilities.

7. The format of the manuscript is chaotic. It is recommended that authors make adjustments according to the submission guidelines.

8. The picture pixels in the manuscript are too low.

Some literature to consult:

Influence of urban morphological characteristics on thermal environment, Sustainable Cities and Society (2021), https://doi.org/10.1016/j.scs.2021.103045.

Coupling Coordination Relationships between Urban-industrial Land Use Efficiency and Accessibility of Highway Networks: Evidence from Beijing-Tianjin-Hebei Urban Agglomeration, China. Sustainability 2019, 11, 1446. https://doi.org/10.3390/su11051446

Understanding land surface temperature impact factors based on local climate zones, Sustainable Cities and Society (2021), doi: https://doi.org/10.1016/j.scs.2021.102818.

Demand prediction and regulation zoning of urban-industrial land: Evidence from Beijing-Tianjin-Hebei Urban Agglomeration, China. Environ Monit Assess 191, 412 (2019). https://doi.org/10.1007/s10661-019-7547-4

Reviewer 2

The work "LANDSCAPE COMPLEXITY METRICS BASED ON INFORMATION ENTROPY" calculates and spatially expresses the complexity of landscape indicators for different regions of interest (ROIs) selected in the satellite image based on the scripts CompPlex HeROI and CompPlex Janus, aiming at revealing the impact of land use change and its fragmentation on landscape complexity. In my opinion, the work is interesting and worthy of publication, however, it requires some clarifications.

1.In the "Introduction" section, "information entropy" is simply mentioned, but as the main content of the research, the background and current related researches of information entropy should be added.

2.In the "Material and Methods" section, López-Ruiz, Mancini, and Calbet (LMC)和Shiner, Davison, and Landsberg (SDL), as key measures, have not been explained clearly, and need to be further clarified.

3.The names of ROIs are inconsistent in the text, Figure 7, Table 2 and Table 3, so it is suggested to unify them.

4.In Table 2 and Table3, the meanings of different colors are best explained in the legend.

5.The clarity of pictures in the manuscript needs to be further improved.

6.Results and discussion are expressed together in the manuscript, but the content expressed is mostly the description of the results, which lacks in-depth discussion. It is suggested that the discussion be set up as a separate part to discuss some key issues, such as:

(1) What are the reasons for the differences in the complexity of different landscapes? How changes in land use and the consequent fragmentation affect the complexity of the landscape?

(2) The manuscript mentioned that“the complexity algorithms are robust”, but how to prove it?

(3) What are the limitations of the research?

Reviewers' comments:

Reviewer's Responses to Questions

**Comments to the Author**

1. Is the manuscript technically sound, and do the data support the conclusions?

Reviewer #1: Yes

Reviewer #2: Partly

2. Has the statistical analysis been performed appropriately and rigorously? 

Reviewer #1: Yes

Reviewer #2: Yes

3. Have the authors made all data underlying the findings in their manuscript fully available?

Reviewer #1: Yes

Reviewer #2: Yes

4. Is the manuscript presented in an intelligible fashion and written in standard English?

Reviewer #1: Yes

Reviewer #2: Yes

5. Review Comments to the Author

Reviewer #1: 1. The title needs to be more detailed.

2. Please highlight the key points in the abstract.

3. The introduction should further highlight what is the motivation of the paper.

4. Material and Methods: 2.2 should be "Methods", "Application of complexity measures based on information entropy of

remote sensing images" etc. should be 2.2.1.

5. Results and discussion: The results should be better described, discussed and justified using references.

6. Conclusion mostly looks like a summary of the work done and the results obtained. No interpretation of the results in given as well as no recommendation for the government and policy makers as to how the results could be used. Also, there should be some content in the conclusion regarding the limitations of the current research and future work possibilities.

7. The format of the manuscript is chaotic. It is recommended that authors make adjustments according to the submission guidelines.

8. The picture pixels in the manuscript are too low.

Some literature to consult:

Influence of urban morphological characteristics on thermal environment, Sustainable Cities and Society (2021), https://doi.org/10.1016/j.scs.2021.103045.

Coupling Coordination Relationships between Urban-industrial Land Use Efficiency and Accessibility of Highway Networks: Evidence from Beijing-Tianjin-Hebei Urban Agglomeration, China. Sustainability 2019, 11, 1446. https://doi.org/10.3390/su11051446

Understanding land surface temperature impact factors based on local climate zones, Sustainable Cities and Society (2021), doi: https://doi.org/10.1016/j.scs.2021.102818.

Demand prediction and regulation zoning of urban-industrial land: Evidence from Beijing-Tianjin-Hebei Urban Agglomeration, China. Environ Monit Assess 191, 412 (2019). https://doi.org/10.1007/s10661-019-7547-4

Reviewer #2: The work "LANDSCAPE COMPLEXITY METRICS BASED ON INFORMATION ENTROPY" calculates and spatially expresses the complexity of landscape indicators for different regions of interest (ROIs) selected in the satellite image based on the scripts CompPlex HeROI and CompPlex Janus, aiming at revealing the impact of land use change and its fragmentation on landscape complexity. In my opinion, the work is interesting and worthy of publication, however, it requires some clarifications.

1.In the "Introduction" section, "information entropy" is simply mentioned, but as the main content of the research, the background and current related researches of information entropy should be added.

2.In the "Material and Methods" section, López-Ruiz, Mancini, and Calbet (LMC)和Shiner, Davison, and Landsberg (SDL), as key measures, have not been explained clearly, and need to be further clarified.

3.The names of ROIs are inconsistent in the text, Figure 7, Table 2 and Table 3, so it is suggested to unify them.

4.In Table 2 and Table3, the meanings of different colors are best explained in the legend.

5.The clarity of pictures in the manuscript needs to be further improved.

6.Results and discussion are expressed together in the manuscript, but the content expressed is mostly the description of the results, which lacks in-depth discussion. It is suggested that the discussion be set up as a separate part to discuss some key issues, such as:

(1) What are the reasons for the differences in the complexity of different landscapes? How changes in land use and the consequent fragmentation affect the complexity of the landscape?

(2) The manuscript mentioned that“the complexity algorithms are robust”, but how to prove it?

(3) What are the limitations of the research?

6. PLOS authors have the option to publish the peer review history of their article (what does this mean?). If published, this will include your full peer review and any attached files.

Reviewer #1: No

Reviewer #2: No

---

## [Author Response · Author response to Decision Letter 0]

24 Dec 2021

In face of PLOS ONE's decision for the manuscript "LANDSCAPE COMPLEXITY METRICS BASED ON INFORMATION ENTROPY" ("Decision: Revision required [PONE-D-21-31069] -[EMID:f31842f6a90d3a9c]"), we thank the editor’s and reviwers’ comments that has been so constructive to elaborate this new version of the submitted paper.

All the comments were considered providing the following changes to the points raised by the academic editor and reviewers:

- “Journal Requirements: “1. Please ensure that your manuscript meets PLOS ONE's style requirements, including those for file naming.” and “Reviewer 1: 7. The format of the manuscript is chaotic. It is recommended that authors make adjustments according to the submission guidelines.”: 

Action(s) performed: We made changes to the new version of the manuscript according to the submission guidelines and PLOS ONE's style requirements, including those for file naming.

- “Reviewer 1: 1. The title needs to be more detailed.”: 

Action(s) performed: We changed the title to be more detailed (new title: ‘METRICS BASED ON INFORMATION ENTROPY APPLIED TO EVALUATE COMPLEXITY OF LANDSCAPE PATTERNS’)

- “Reviewer 1: 2. Please highlight the key points in the abstract.”: 

Action(s) performed: We added a sentence to the Abstract highlighting the main results obtained (“As expected, both for the complexity patterns evaluated by CompPlex HeROI and the complexity maps generated by CompPlex Janus, the areas with vegetation located in a region of intermediate spatial heterogeneity had lower values for the He and He/Hmax measures and higher values for the LMC and SDL measurements.”).

- “Reviewer 1: 3. The introduction should further highlight what is the motivation of the paper.” and “Reviewer 2: 1. In the "Introduction" section, "information entropy" is simply mentioned, but as the main content of the research, the background and current related researches of information entropy should be added.”: 

Action(s) performed: We include in the 8th paragraph of the 'Introduction' background and more current related researches of information entropy (references 25-35). 

We have also modified the 9th (last) paragraph of the 'Introduction' to give more emphasis to our objective ("From the case study presented, this article aims to show how such measures can be used to evaluate and indicate how changes in land use and fragmentation affect the complexity of the landscape, and consequently, to indicate how measures based on entropy information can be used as indicators of its integrity and resilience.")

- “Journal Requirements: 2. In your Methods section, please provide additional location information, including geographic coordinates of your field collection site if available.”:

Action(s) performed: We add to the text the geographic coordinates of the Itirapina Ecological and Experimental Parks

- “Reviewer 1: 8. The picture pixels in the manuscript are too low.” and “Reviewer 2: 5.The clarity of pictures in the manuscript needs to be further improved.”:

Action(s) performed: We improved the quality of the images.

- “Reviewer 2: 2. In the "Material and Methods" section, López-Ruiz, Mancini, and Calbet (LMC) and Shiner, Davison, and Landsberg(SDL), as key measures, have not been explained clearly, and need to be further clarified.”: ????

Action(s) performed: In item "2.2.1) Application of complexity measures based on information entropy of remote sensing images", we added two paragraphs (4th and 5th paragraphs) explaining in more detail about the LMC and SDL measures.

- “Journal Requirements: 3. We note that Figures 1 and 7 in your submission contain [map/satellite] images which may be copyrighted. All PLOS content is published under the Creative Commons Attribution License (CC BY 4.0), which means that the manuscript, images, and Supporting Information files will be freely available online, and any third party is permitted to access, download, copy, distribute, and use these materials in any way, even commercially, with proper attribution. For these reasons, we cannot publish previously copyrighted maps or satellite images created using proprietary data, such as Google software (Google Maps, Street View, and Earth). For more information, see our copyright guidelines: http://journals.plos.org/plosone/s/licenses-and-copyright. We require you to either (1) present written permission from the copyright holder to publish these figures specifically under the CC BY 4.0 license, or (2) remove the figures from your submission: a. You may seek permission from the original copyright holder of Figures 1 and 7 to publish the content specifically under the CC BY 4.0 license. We recommend that you contact the original copyright holder with the Content Permission Form(http://journals.plos.org/plosone/s/file?id=7c09/content-permission-form.pdf) and the following text:“I request permission for the open-access journal PLOS ONE to publish XXX under the Creative Commons Attribution License (CCAL) CC BY 4.0 (http://creativecommons.org/licenses/by/4.0/). Please be aware that this license allows unrestricted use and distribution, even commercially, by third parties. Please reply and provide explicit written permission to publish XXX under a CC BY license and complete the attached form.” Please upload the completed Content Permission Form or other proof of granted permissions as an "Other" file with your submission. In the figure caption of the copyrighted figure, please include the following text: “Reprinted from [ref] under a CC BY license, with permission from [name of publisher], original copyright [original copyright year].” b. If you are unable to obtain permission from the original copyright holder to publish these figures under the CC BY4.0 license or if the copyright holder’s requirements are incompatible with the CC BY 4.0 license, please either i) remove the figure or ii) supply a replacement figure that complies with the CC BY 4.0 license. Please check copyright information on all replacement figures and update the figure caption with source information. If applicable, please specify in the figure caption text when a figure is similar but not identical to the original image and is therefore for illustrative purposes only. The following resources for replacing copyrighted map figures may be helpful: USGS National Map Viewer (public domain): http://viewer.nationalmap.gov/viewer/The Gateway to Astronaut Photography of Earth (public domain): http://eol.jsc.nasa.gov/sseop/clickmap/Maps at the CIA (public domain): https://www.cia.gov/library/publications/the-world-factbook/index.html andhttps://www.cia.gov/library/publications/cia-maps-publications/index.htmlNASA Earth Observatory (public domain):http://earthobservatory.nasa.gov/Landsat: http://landsat.visibleearth.nasa.gov/USGS EROS (Earth Resources Observatory and Science (EROS) Center) (public domain): http://eros.usgs.gov/#Natural Earth (public domain): http://www.naturalearthdata.com/”: 

Action(s) performed: For Figure 1 we replaced the Google Earth satellite image with a Sentinel-2 (ESA) image obtained from the U.S. Geological Survey, which complies with the CC BY 4.0 license. In turn, we have kept Figure 7 of the CBERS-4 satellite image (which also complies with the CC BY 4.0 license) and added the source of information in the figure caption.

- “Reviewer 1: 5. Results and discussion: The results should be better described, discussed and justified using references” and “Reviewer 2: 6. Results and discussion are expressed together in the manuscript, but the content expressed is mostly the description of the results, which lacks in-depth discussion. It is suggested that the discussion be set up as a separate part to discuss some key issues, such as:(1) What are the reasons for the differences in the complexity of different landscapes? How changes in landuse and the consequent fragmentation affect the complexity of the landscape? (2) The manuscript mentioned that “the complexity algorithms are robust”, https://yout.com/youtube-mp3/?lang=ptbut how to prove it?(3) What are the limitations of the research?”: ????

Action(s) performed: We seek to describe, discuss and justify the results by giving more details about them, explaining their meanings and using references to support our justifications. Therefore, we chose to leave the discussions together with the results and, in this way, better clarify the three questions raised by the second reviewer.

- “Reviewer 2: 3. The names of ROIs are inconsistent in the text, Figure 7, Table 2 and Table 3, so it is suggested to unify them.” and “Reviewer 2: 4.In Table 2 and Table3, the meanings of different colors are best explained in the legend.”: 

Action(s) performed: We unify the names of ROIs in the text, Figure 7 and Tables 1-3.

- “Reviewer 1: 6. Conclusion mostly looks like a summary of the work done and the results obtained. No interpretation of the results in given as well as no recommendation for the government and policy makers as to how the results could be used. Also, there should be some content in the conclusion regarding the limitations of the current research and future work possibilities.”:

 Action(s) performed: We modified the text of the conclusions, seeking to highlight the most important results achieved and their relevance for research in Landscape Ecology and for its use in environmental planning and management. We also added issues related to possible limitations of the measures used, as well as the possibility of future studies in which they can be used.

- “Reviewer 1: Some literature to consult: 

- Influence of urban morphological characteristics on thermal environment, Sustainable Cities and Society (2021),https://doi.org/10.1016/j.scs.2021.103045.

- Coupling Coordination Relationships between Urban-industrial Land Use Efficiency and Accessibility of Highway Networks: Evidence from Beijing-Tianjin-Hebei Urban Agglomeration, China. Sustainability 2019, 11, 1446.https://doi.org/10.3390/su11051446

- Understanding land surface temperature impact factors based on local climate zones, Sustainable Cities and Society(2021), doi: https://doi.org/10.1016/j.scs.2021.102818.

- Demand prediction and regulation zoning of urban-industrial land: Evidence from Beijing-Tianjin-Hebei Urban Agglomeration, China. Environ Monit Assess 191, 412 (2019). https://doi.org/10.1007/s10661-019-7547-4”

Action(s) performed: We appreciate the literature suggestions, which were included in our text.

Sincerely yours

The authors

---

## [Decision Letter · Decision Letter 1]

3 Jan 2022

Metrics based on information entropy applied to evaluate complexity of landscape patterns

PONE-D-21-31069R1

Dear Dr. Mattos,

We’re pleased to inform you that your manuscript has been judged scientifically suitable for publication and will be formally accepted for publication once it meets all outstanding technical requirements.

Kind regards,

Jun Yang

Academic Editor

PLOS ONE

Additional Editor Comments (optional):

Accept

Reviewers' comments:

Reviewer's Responses to Questions

**Comments to the Author**

1. If the authors have adequately addressed your comments raised in a previous round of review and you feel that this manuscript is now acceptable for publication, you may indicate that here to bypass the “Comments to the Author” section, enter your conflict of interest statement in the “Confidential to Editor” section, and submit your "Accept" recommendation.

Reviewer #1: (No Response)

Reviewer #2: All comments have been addressed

2. Is the manuscript technically sound, and do the data support the conclusions?

Reviewer #1: (No Response)

Reviewer #2: Yes

3. Has the statistical analysis been performed appropriately and rigorously? 

Reviewer #1: (No Response)

Reviewer #2: Yes

4. Have the authors made all data underlying the findings in their manuscript fully available?

Reviewer #1: (No Response)

Reviewer #2: Yes

5. Is the manuscript presented in an intelligible fashion and written in standard English?

Reviewer #1: (No Response)

Reviewer #2: Yes

6. Review Comments to the Author

Reviewer #1: The authors have adequately addressed the comments raised in a previous round of review and I feel that this manuscript is now acceptable for publication.

Reviewer #2: An interesting and valuable study paper, the authors have made progressive revisions, I do not have much too many comments about the study. The paper is suitable for publication after minor revision on grammar mistakes.

7. PLOS authors have the option to publish the peer review history of their article (what does this mean?). If published, this will include your full peer review and any attached files.

Reviewer #1: No

Reviewer #2: No

---

## [Editor Report · Acceptance letter]

6 Jan 2022

PONE-D-21-31069R1 

Metrics based on information entropy applied to evaluate complexity of landscape patterns 

Dear Dr. Mattos:

I'm pleased to inform you that your manuscript has been deemed suitable for publication in PLOS ONE. Congratulations! Your manuscript is now with our production department. 

Kind regards, 

on behalf of

Dr. Jun Yang 

Academic Editor

PLOS ONE